# Outdoor Absolute Humidity Predicts the Start of Norovirus GII Epidemics

Thomas Beck-Friis,[a,b] Nicklas Sundell,[a,b] Lars Gustavsson,[a,b] Magnus Lindh,[a,c] Johan Westin,[a,b] Lars-Magnus Andersson[a,b]

[a]Department of Infectious Diseases, Sahlgrenska University Hospital, Region Västra Götaland, Gothenburg, Sweden
[b]Department of Infectious Diseases, Institute of Biomedicine, University of Gothenburg, Gothenburg, Sweden
[c]Department of Microbiology, Institute of Biomedicine, University of Gothenburg, Gothenburg, Sweden

**ABSTRACT**  Seasonal variation of viral gastroenteritis is related to weather conditions, but the relationship with the incidence of viral gastroenteritis (GE) is not fully understood. This study examined the impact of outdoor climate factors on seasonal variation in detection rates of gastroenteritis viruses, with emphasis on norovirus. Weekly detection rates of norovirus genogroup I (GI) and II (GII), rotavirus, adenovirus, astrovirus, and sapovirus were analyzed in relation to average weekly means of meteorological parameters. Associations between rates of PCR detection of the viral GE pathogens and climate factors were investigated with generalized linear models. Low absolute humidity was correlated with increased detection of adenovirus ($P = 0.007$), astrovirus ($P = 0.005$), rotavirus ($P = 0.004$), norovirus GI ($P = 0.001$), and sapovirus ($P = 0.002$). In each investigated season, a drop in absolute humidity preceded the increase in norovirus GII detections. We found a correlation between declining absolute humidity and increasing norovirus GII detection rate. Absolute humidity was a better predictor of gastrointestinal virus seasonality compared to relative humidity.

**IMPORTANCE**  Viral gastroenteritis causes considerable morbidity, especially in vulnerable groups such as the elderly and chronically ill. Predicting the beginning of seasonal epidemics is important for the health care system to withstand increasing demands. In this paper we studied the association of outdoor climate factors on the detection rates of gastrointestinal viruses and the association between these factors and the onset of annual norovirus epidemics. Declining absolute humidity preceded the increase in diagnosed norovirus GII cases by approximately 1 week. These findings contribute to the understanding of norovirus epidemiology and allow health care services to install timely preventive measures and can help the public avoid transmission.

**KEYWORDS**  norovirus, rotavirus, weather/epidemiology, humidity, gastroenteritis/epidemiology

Acute viral gastroenteritis is a major cause of morbidity and mortality worldwide (1). Mortality due to viral GE is low in developed countries, but viral GE still accounts for a considerable morbidity and burden of disease (2).

In the temperate climate zone of the Northern hemisphere, norovirus GII and rotavirus epidemics peak during the winter season. The onset of the norovirus GII seasonal epidemics may differ several weeks or even months from year to year, while rotavirus epidemics most often peak during late winter or early spring (3).

Even though thoroughly studied, the mechanisms behind seasonality of viral GE infections remain unknown to a large extent. Outdoor climate factors, however, seem to be of importance to the seasonality of viral respiratory infections as well as viral GE infections (3–5).

Previous studies have indicated that low outdoor temperature is associated with annual norovirus and rotavirus epidemics in temperate climates (3, 5–7). There are, however, contradicting studies from temperate climates in the Southern Hemisphere where norovirus

Address correspondence to Thomas Beck-Friis, thomas.andreasson@gu.se.

The authors declare no conflict of interest.

*[This article was published on 14 February 2023 with the incorrect corresponding author. The corresponding author information was corrected in the current version, posted on 27 February 2023.]*

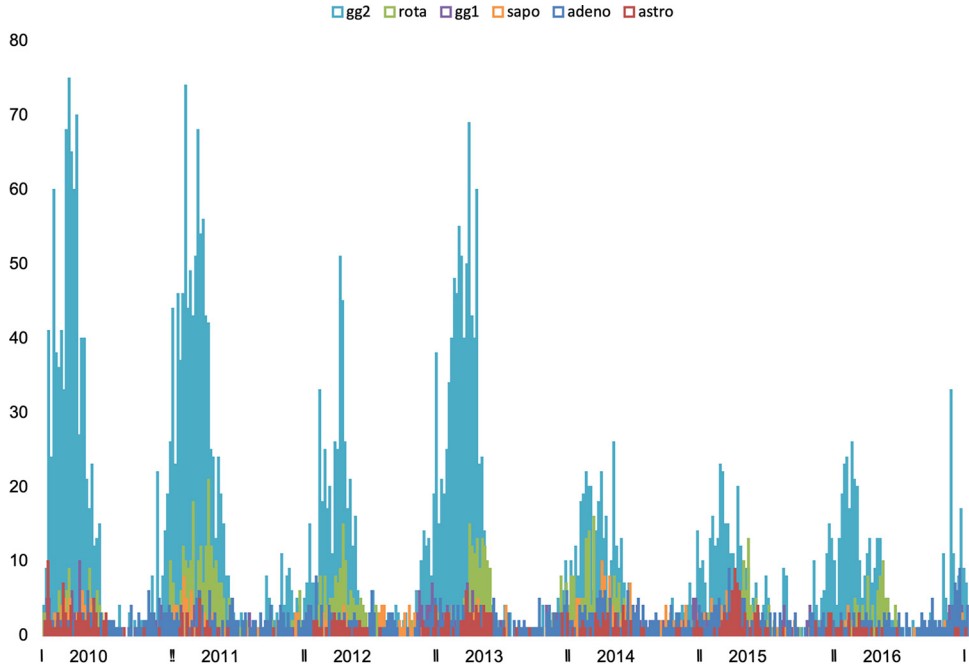

**FIG 1** Weekly incidence of various gastrointestinal viruses, detected with real-time PCR, according to season across seven consecutive years 2010–2016 (n = 7337). All agents except norovirus GI and adenovirus have a strong seasonal pattern with winter epidemics and almost no occurrence in the summer months. gg2 = norovirus GII (turquoise), rota = rotavirus (green), gg1 = norovirus GI (purple), sapo = sapovirus (orange), adeno = adenovirus (blue) and astro = astrovirus (red).

epidemics occur during the warmer months (8, 9). As temperature alone cannot explain the association of outdoor climate factors on viral transmission, there has been a shift of focus toward the role of humidity. Low relative humidity (RH; the water content in a gas, relative to the maximum capacity of water vapor that gas can hold, at a given temperature) has been associated with increased transmission and viability of both norovirus and rotavirus (3, 10, 11). In a temperate climate, however, outdoor RH is peaking during winter. Absolute humidity (AH; the amount of water per volume unit of air [g/m³], irrespective of temperature), may be a better predictor of the seasonal variation of influenza and norovirus detection rate rather than RH (11–13).

The aim of the study was to study the impact of outdoor climate factors on the detection rates of GE viruses in a large clinical data set, and the association between these factors and the onset of annual norovirus epidemics in particular.

## RESULTS

A total of 24,630 samples were analyzed, of which 7,337 (30%) were positive for at least one viral GE agent. Norovirus genogroup II (GII) was the most common agent detected, accounting for 4,447 (61%) of the positive samples.

**Seasonality of GE epidemics.** Norovirus GII, sapovirus, rotavirus, and astrovirus followed a distinct seasonal pattern with a peak during the winter season, while norovirus genogroup I (GI) and adenovirus were detected across all seasons (Fig. 1).

Weekly AH, RH, VP, precipitation, and temperature all followed seasonal patterns ($p < 0.001$), while wind speed did not ($P = 0.7$). The seasonal patterns of absolute and relative humidity are displayed in relation to norovirus GII and rotavirus in Fig. 2 and 3, respectively.

Univariate analyses comparing weekly detection rate of each virus with meteorological factors showed that low absolute humidity and low vapor pressure correlated with detection of all viruses, except norovirus GII (Table 1). Low temperature correlated with elevated detection rates of all viruses except norovirus GII and rotavirus. In addition, a significant correlation was seen between low precipitation and sapovirus detection rates ($P = 0.01$).

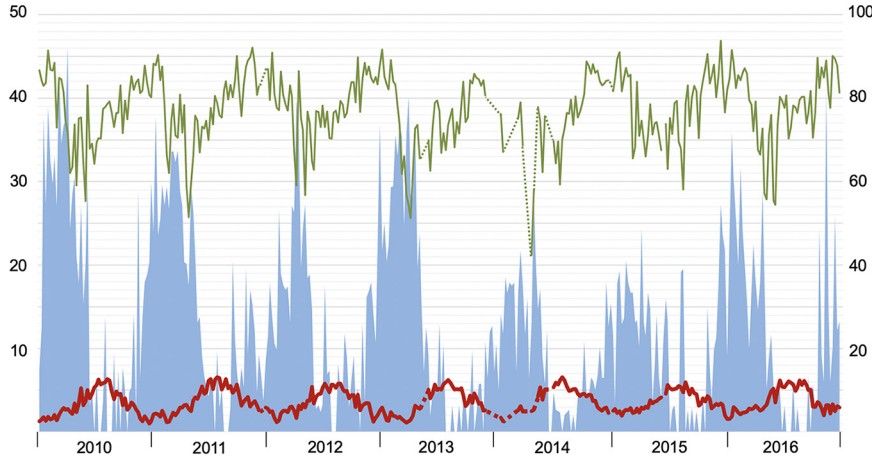

**FIG 2** Weekly positivity rate of norovirus genogroup II infections (% left axis; blue bars) according to average weekly outdoor absolute humidity (g $H_2O/m^3$ right axis; thick red line) and relative humidity (% right axis; thin green line) across the 7-year study period (2010-2016). Dashed lines represent gaps between weeks with missing AH and RH data.

**Impact of humidity on beginning of norovirus epidemics.** Every season, the start of the norovirus epidemic coincided with a drop in AH, and declining detection rates coincided with a distinct rise in AH. To study these correlations in further detail, we analyzed weekly norovirus positivity rates during the incline (Fig. 4) and decline period (Fig. 5) of every season. When studying the inclination of each yearly epidemic, correlations were observed for AH, VP, temperature, and RH (Table 2).

The strongest correlations between outdoor climate factors and weekly positivity rates were seen when lagging weather factors. AH and temperature showed the strongest effect (F value) on norovirus GII when lagging weather data set 1 week, corresponding to a 1-week period between a drop in outside AH and temperature before reaching the steepest incline of the seasonal norovirus epidemic (Supplementary table I and II). When assessing rotavirus using the same lagging strategy the strongest correlation was seen at 6- and 7-week lags for AH and temperature, respectively (Supplementary table IV and V).

We observed longer lag time optimums when comparing norovirus and rotavirus to RH: 12 and 14 weeks, respectively (Table S3 and 4). AH had more prominent lag time optimum correlations with norovirus GII and rotavirus than RH (F value: 25 and 97 versus 7 and 10, respectively).

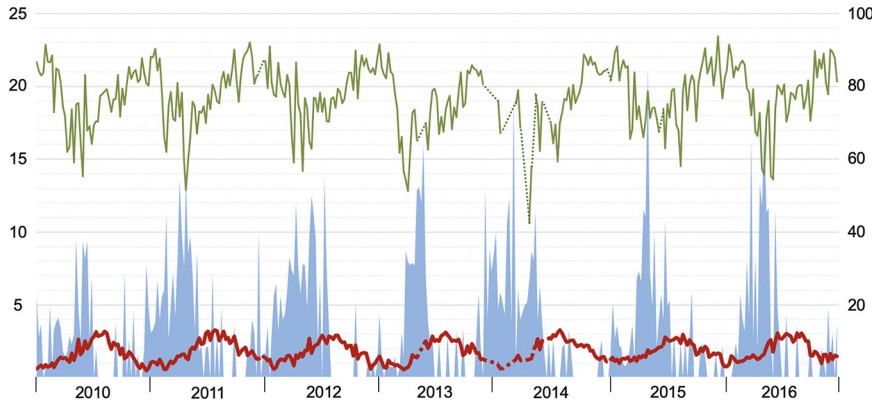

**FIG 3** Weekly positivity rate of rotavirus infections (% left axis; blue bars) according to average weekly outdoor absolute humidity (g $H_2O/m$ right axis[3]; thick red line) and relative humidity (% right axis; thin green line) across the 7-year study period (2010–2016). Dashed lines represent gaps between weeks with missing AH and RH data.

**TABLE 1** Correlations between gastrointestinal viruses and weather factors during the 7-year study period, 2010–2016[a]

| Virus | Temp | | Relative humidity | | Vapor pressure | | Absolute humidity | | Wind speed | | Precipitation | |
|---|---|---|---|---|---|---|---|---|---|---|---|---|
| | Coeff | p | Coeff | p | Coeff | p | Coeff | p | Coeff | p | Coeff | p |
| Adenovirus | −0.02 | 0.02[b] | 0.003 | 0.6 | $-4 \times 10^{-4}$ | 0.008[b] | −0.05 | 0.007[b] | 0.04 | 0.3 | 0.001 | 0.5 |
| Astrovirus | −0.05 | 0.02[b] | 0.003 | 0.8 | −0.001 | 0.005* | −0.2 | 0.005[b] | 0.1 | 0.09 | −0.007 | 0.2 |
| Rotavirus | −0.03 | 0.1 | −0.01 | 0.1 | $-8 \times 10^{-4}$ | 0.004* | −0.1 | 0.004[b] | 0.08 | 0.2 | −0.003 | 0.4 |
| Norovirus GI | −0.07 | $3 \times 10^{-4b}$ | 0.009 | 0.5 | −0.001 | 0.001* | −0.2 | 0.001[b] | 0.04 | 0.6 | −0.003 | 0.6 |
| Norovirus GII | −0.008 | 0.5 | 0.008 | 0.1 | $-4 \times 10^{-4}$ | 0.1 | −0.05 | 0.2 | 0.03 | 0.5 | $5 \times 10^{-4}$ | 0.8 |
| Sapovirus | −0.04 | 0.02[b] | −0.002 | 0.8 | $-9 \times 10^{-4}$ | 0.002* | −0.1 | 0.002[b] | −0.06 | 0.3 | −0.009 | 0.01[b] |

[a]Coeff = Coefficient, p = P value.
[b]Significant P value.

## DISCUSSION

We show that a drop in outside absolute humidity and temperature precede the rise of seasonal norovirus and rotavirus detection rates.

Detection rates of viral GE agents showed a stronger correlation with AH compared to RH, which is consistent with findings in viral respiratory infections such as influenza (12). Previous studies have mostly focused on temperature, precipitation, and RH to describe the relationship between outdoor climate factors and incidence of viral respiratory and GE infections, but recent publications indicate that outdoor AH may be more relevant than outdoor RH to predict a seasonal rise in influenza (12, 13) and norovirus (11) infections. In agreement with our findings, Atchison et al. showed a strong inverse correlation between temperature and rotavirus detection, and a weaker inverse correlation between RH and rotavirus detection (14). Though, in contrast to our findings, they found no associations between AH and rotavirus detection.

Several mechanisms can contribute to the increased spread of norovirus in a setting with low AH as seen in our study. Colas de la Noue et al. have shown that low AH favored murine norovirus survival and infectivity (and norovirus VLP stability) in the environment (11). Evaporation of aerosolized droplets has been suggested as a possible mechanism (10, 15, 16), since small droplets, after vomiting, may remain aerosolized for a longer time and travel further distance when AH is low. The combination of larger contaminated areas and increased viral survival and stability on surfaces could affect transmission via the fecal-oral route and contribute to the annual increase in norovirus infections during wintertime in a temperate climate.

Other factors, such as antigenic variation (3), declining herd immunity (3, 5), long-term shedding (17), and social behavior such as indoor crowding may also contribute to the seasonal variation of norovirus as well as other viral pathogens.

We found that norovirus GII, rotavirus, sapovirus, and astrovirus all showed distinct seasonal patterns of detection with peaks during the winter or early spring season, whereas

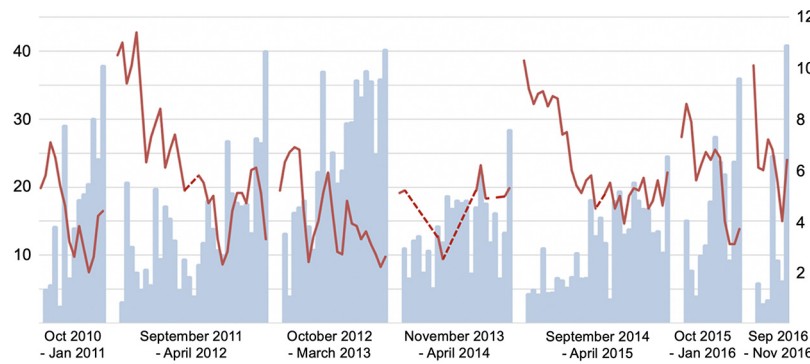

**FIG 4** Weekly positivity rate of norovirus GII positives from the last week of no positive samples to peak positivity rate (% left axis; blue bars) in relation to outdoor absolute humidity (g $H_2O/m^3$ right axis; red lines) across seven seasons (2010–2016). Dashed lines represent gaps between weeks with missing AH data.

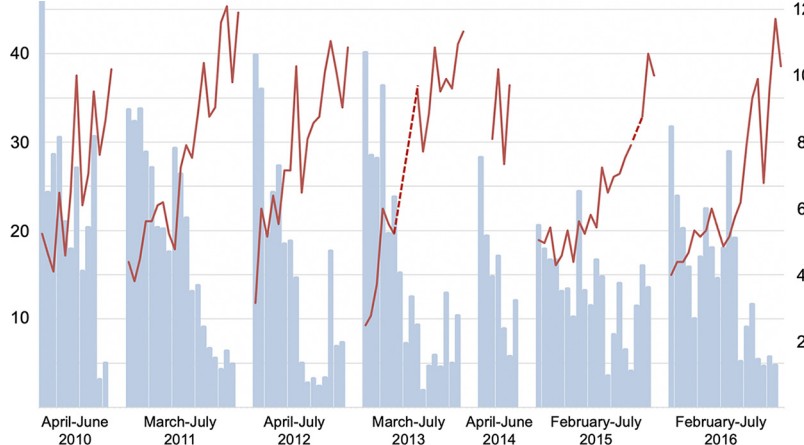

**FIG 5** Weekly positivity rate of norovirus GII positives from peak positivity rate to the last week of no positive samples (% left axis; blue bars) in relation to outdoor absolute humidity (g $H_2O/m^3$ right axis; red lines) across seven seasons (2010–2016). Dashed lines represent gaps between weeks with missing AH data.

norovirus GI and adenovirus were more evenly distributed across all seasons. For adenovirus, the lack of seasonality is in line with previous studies (18, 19). Norovirus GI is associated with water- and foodborne outbreaks and has less potential for person-to-person spread, which could explain why no distinct seasonality was observed (20).

Absolute humidity was the better predictor among the outdoor weather factors.

As transmission of viruses often occurs indoors, outdoor climate factors cannot fully explain seasonal variations in detection rates of viral infections. It has previously been suggested that mass-based outdoor humidity, such as AH and specific humidity, serves as the best proxy for indoor climate conditions (21). Unlike RH or temperature, AH shows a high correlation between indoor and outdoor settings. It might therefore be a more useful predictor of viral transmission, especially since it is not a temperature-dependent ratio (12, 21–23). It has also been suggested that AH should be preferred over RH in medical research (24).

Absolute humidity correlates well with virus transmission, although the mechanisms are difficult to explain. In an attempt to explain the relationship between virus viability and climate factors it may be preferable to study AH as a function of RH and temperature, using the ideal gas law. RH affects droplet size and eventual rupture, while increased temperature inactivates exposed nucleic acids and proteins, which may explain the combined effects of AH on virus viability (25).

The strongest correlations between outdoor climate factors and viral positivity rate were seen when lagging outdoor climate factors one to several weeks, which is in line with previous reports (14, 26). In our study, norovirus reached peak inclination several weeks faster than rotavirus, when counting from the largest drop in AH (and temperature). The observation that rotavirus epidemics develop more slowly, compared to norovirus epidemics, under the influence of outdoor climate factors is in agreement with previous observations (26). Numerous factors likely contribute to differing lag times for each

**TABLE 2** Correlations between norovirus genogroup II and weather factors during the first weeks of each season's epidemic, 2010 to 2016

| Weather factor (weekly mean) | Fixed effects F | Coefficient | P value |
|---|---|---|---|
| Temp | 14 | −0.07 | 0.0002[a] |
| Vapor pressure | 31 | −0.002 | $1.5 \times 10^{-7}$[a] |
| Relative humidity | 11 | −0.03 | 0.01[a] |
| Absolute humidity | 32 | −0.3 | $8.7 \times 10^{-8}$[a] |
| Wind speed | 0.08 | 0.02 | 0.78 |
| Precipitation | 0.04 | −0.005 | 0.85 |

[a]Significant P value.

virus. Norovirus GII has a shorter incubation period and a larger proportion of asymptomatic carriers than rotavirus (27), and the forceful vomiting typical for norovirus disease contributes to efficient spread. Furthermore, adolescents and adults are susceptible to norovirus since immunity is genotype-, or even strain-specific and not as long-lasting as immunity to rotavirus, generating a much larger susceptible population (28).

This study has several limitations. First, we have not been able to adjust for confounders such as herd immunity, viral evolution, and emergence of new viral strains, or behavioral factors such as school holidays and indoor crowding, which could affect the results.

Second, we only included a limited number of outdoor climate factors. Other factors, such as UV radiation, could also affect viral transmission (29).

Finally, this study was performed using data from one meteorological station and a single clinical center and our observations may not be applicable to other regions than our geographical setting.

Additional studies are needed to illuminate the impact of humidity and other climate factors on GE virus transmission. As transmission mainly occurs indoors, it would be of interest to study indoor climate factors in relation to incidence of viral gastroenteritis.

In conclusion, we found a correlation between declining absolute humidity and increasing norovirus GII and rotavirus detection rates. Absolute humidity was a better predictor of detection rates of norovirus GII and rotavirus than relative humidity. A drop in absolute humidity preceded the increase in norovirus GII positivity rate each season. Peak effect was reached 1 week after the drop in absolute humidity.

## MATERIALS AND METHODS

**Study design and detection of GE agents.** Sahlgrenska University Hospital is the only hospital serving the city of Gothenburg with a population of approximately 680,000 inhabitants. All clinical samples referred to the hospital during the 7-year study period (from January 2010 through December 2016), for standard of care testing by routine multiplex PCR for viral agents causing GE, were included retrospectively. There was no change in standard of care testing over the course of the study period. All age groups were represented and included in the analysis. Samples were predominantly from hospitalized patients, but also from outpatient clinics and primary care settings to a minor extent. No clinical patient data were available.

Samples were either stool or vomit samples, or rectal swab samples collected with flocked swabs. As previously described, a multiplex PCR panel was used for detection of six viral gastrointestinal pathogens (adenovirus, astrovirus, norovirus GI and GII, rotavirus, and sapovirus) (30).

**Meteorological data.** Data on average weekly outdoor temperature (degrees Celsius), vapor pressure (VP; Pa), absolute humidity (AH; $gH_2O/m^3$), relative humidity (RH; %), wind speed (m/s), and precipitation (mm) for the study period were obtained from the Swedish Meteorological and Hydrological Institute's local weather station in Gothenburg (5 m above sea level, situated at Latitude: 57.7157N, Longitude: 11.9925E).

Detection rates (weekly number of positive samples) of viral agents included in the standard of care PCR panel and the weekly means of included meteorological parameters were used to analyze seasonality and correlations. A total of 365 paired, consecutive weekly observations were included.

In order to study weather conditions during the incline of norovirus GII epidemics, subsets of weekly data were extracted. The viral incline period was defined as the last week with no positive samples to the week of peak positivity rate.

Delayed climate effects on GE virus detection rates were investigated by introducing a lag model in which data from outdoor climate factors were lagged week by week to identify the optimum threshold.

**Statistical methods.** As the underlying data are counts with a temporal dependency, we chose to use a generalized linear mixed model (GLMM) based on the Poisson distribution, including time series components. When fitting the models without accounting for time dependencies the residuals were autocorrelated, especially for norovirus GII. Adding a single autoregressive term removed the autocorrelation in the residuals. The auto.arima function in the "Forecast" R-package selected models containing a mix of autoregressive and moving average terms to model the data. When adding weather factors as regressors models with only autoregressive terms were suggested, indicating that a single autoregressive term, e.g., an AR1 model, would be sufficient to account for the temporal dependencies.

In addition, for norovirus GII, data subsets from the first weeks of each yearly epidemic were analyzed. Delayed weather effects on norovirus and rotavirus detection rates were analyzed using a lag model. Optimum thresholds were identified by repeated GLMM analyses, lagging climate factors 1 week at a time, until each model's greatest F value was determined. For example, the effect of 'lag one' represented the effect of the climate factor measured 1 week prior, on the current week's positivity rate. The statistical analysis was made using the SPSS software package version 27.0.0.0 (IBM, Armonk, NY) and R version 4.2.2.

**Ethical approval.** Not required. All samples were collected for clinical purposes. Data were deidentified prior to analysis and no personal information was available.

## SUPPLEMENTAL MATERIAL

Supplemental material is available online only.

**SUPPLEMENTAL FILE 1**, PDF file, 0.1 MB.

## ACKNOWLEDGMENTS

This study was supported by grants from the Gothenburg Medical Society (grant number GLS-935274).

The authors have no conflicts of interest to declare.

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
