## [Reviewer comments · Microbiology Spectrum]

Microbiology Spectrum

Outdoor absolute humidity predicts the start of norovirus GII epidemics.

Thomas Beck-Friis, Nicklas Sundell, Lars Gustavsson, Magnus Lindh, Johan Westin, and Lars-Magnus Andersson

Corresponding Author(s): Thomas Beck-Friis, Biomedicine, Sahlgrenska Academy, University of Gothenburg

Review Timeline:

Submission Date:	June 28, 2022
Editorial Decision:	August 15, 2022
Revision Received:	September 1, 2022
Editorial Decision:	October 18, 2022
Revision Received:	January 15, 2023
Accepted:	January 25, 2023

Editor: Gabriel Parra

Reviewer(s): Disclosure of reviewer identity is with reference to reviewer comments included in decision letter(s). The following individuals involved in review of your submission have agreed to reveal their identity: Jessica Rothman (Reviewer #2)

Transaction Report:

DOI: <https://doi.org/10.1128/spectrum.02433-22>

August 15, 2022

Dr. Thomas Beck-Friis
Biomedicine, Sahlgrenska Academy, University of Gothenburg
Infectious Diseases
Diagnosvägen 21
Gothenburg 41650
Sweden

Re: Spectrum02433-22 (Outdoor absolute humidity predicts seasonal variation of norovirus GII infection.)

Dear Dr. Thomas Beck-Friis:

Thank you for submitting your manuscript to Microbiology Spectrum. Two experts on norovirus epidemiology reviewed your manuscript. While both agree is a subject of interest, one of the reviewers is concerned about the methods of statistical analysis utilized for the data analyzed. Copies of the reviewers' comments are below for your consideration and review.

Link Not Available

Sincerely,

Gabriel Parra

Journals Department
Reviewer comments:

Reviewer #1 (Comments for the Author):

Beck-Friis et al looked at weather conditions as a possible explanation of the seasonality of rotavirus and norovirus GII in Sweden. They plotted the incidence of the 5 gastroenteritis viruses, detected by real-time RT-PCR between 2011 and 2016 in samples from patients of all ages in a single hospital, against the relative and absolute outdoor humidity and found a strong correlation between a drop in the absolute humidity and the increase of norovirus GII and rotavirus.

There are few studies that have looked at the relation between outdoor absolute humidity and start of the norovirus season and this study provides a nice contribution to the literature and would help alert public health departments when to expect the annual start of a virus that often overwhelms healthcare systems.

I have a couple of minor suggested edits:

Line 11: reword 'outdoor climate factors' with 'weather conditions'

Line 12" replace 'between weather conditions and' with 'with'

Line 15" delete 'agents causing viral' and add 'viruses' after 'gastroenteritis'

Line 21: replace 'agents' with 'viruses'

Line 25 AH: spell out in full the first time used

Line 36: would replace 'impact', here and in line 70 with 'association'

Line 88: is the assay specific for enteric (group F) adenoviruses or for all adenoviruses?

Reviewer #2 (Comments for the Author):

The authors propose an interesting topic of investigation. My main concern with this study is the methods of analysis for their data. The use of a linear model (univariate or multiple) is appropriate for an outcome variable that is a proportion (positivity rate) or count (number of cases), and generalized linear model should be used instead. Further, with the use of time-series data the impact of autocorrelation and for meteorological data, seasonality, need to be explored and likely incorporated into the model. There does not appear to be a clear strategy for determining which meteorological variables and lag time lengths for each variable should be included in each of the multivariable models.

Staff Comments:

Preparing Revision Guidelines

Please return the manuscript within 60 days; if you cannot complete the modification within this time period, please contact me. If you do not wish to modify the manuscript and prefer to submit it to another journal, please notify me of your decision immediately so that the manuscript may be formally withdrawn from consideration by Microbiology Spectrum.

Outdoor absolute humidity predicts seasonal variation of norovirus GII infection.

Thomas Beck-Friis^{1,2*}, Nicklas Sundell^{1,2}, Lars Gustavsson^{1,2}, Magnus Lindh^{1,3}, Johan

Westin^{1,2}, Lars-Magnus Andersson^{1,2}

¹Department of Infectious Diseases, Sahlgrenska University Hospital, Region Västra Götaland, Gothenburg,
Sweden

²Department of Infectious Diseases, Institute of Biomedicine, University of Gothenburg, Gothenburg, Sweden

³Department of Microbiology, Institute of Biomedicine, University of Gothenburg, Gothenburg, Sweden

Abstract

*Background:* Seasonal variation of viral gastroenteritis is related to outdoor climate factors,
but the relationship between weather conditions and the incidence of viral gastroenteritis (GE)
is not fully understood.

*Objectives:* To examine the impact of outdoor climate factors on seasonal
variation in detection rates of agents causing viral gastroenteritis, with emphasis on norovirus.

*Study design:* Weekly detection rates of norovirus genogroup I (GI) and II (GII), rotavirus,
adenovirus, astrovirus and sapovirus detection were analyzed in relation to average weekly
means of included meteorological parameters. Associations between rates of PCR detection of
the viral GE pathogens and climate factors were investigated with linear regression models.

*Results:* Low absolute humidity and low temperature were correlated with increased detection
of all agents in the panel, (adjusted R²): adenovirus (0.036; p=0.0002), astrovirus (0.13; p=2 x
10⁻¹²), rotavirus (0.14; p=4 x 10⁻¹³), norovirus GI (0.13; p=4 x 10⁻¹²), norovirus GII (0.36; p=2
23 x 10⁻³⁴), and sapovirus (0.12; p=4 x 10⁻¹¹). Detection of norovirus GII, rotavirus, astrovirus
and sapovirus followed seasonal patterns, while adenovirus and norovirus GI detection did
not. In each investigated season, a sharp drop in AH preceded the increase in norovirus

positivity rate and peak positivity rate was reached four weeks after the drop in absolute
humidity.

*Conclusions:* We found a strong correlation between low AH and norovirus GII positivity
rate. Absolute humidity was a better predictor of positivity rates of norovirus GII and
rotavirus than relative humidity.

**Importance:**

Viral gastroenteritis causes considerable morbidity, especially in vulnerable groups such as
the elderly and chronically ill. Predicting the beginning of seasonal epidemics is important for
the health care system to withstand the increasing demands.

In this paper we studied the impact of outdoor climate factors on the detection rates of
gastrointestinal viruses and the association between these factors and the onset of annual
norovirus epidemics.

A sharp fall in absolute humidity preceded the increase in diagnosed norovirus cases by
approximately four weeks. These findings contribute to the understanding of norovirus
epidemiology and allow health care services to install timely preventive measures and can
help the public avoid transmission.

**Keywords (MeSH):** Norovirus, rotavirus, weather / epidemiology, humidity, gastroenteritis /
epidemiology

**Background**

Acute viral gastroenteritis is a major cause of morbidity and mortality worldwide (1).

Mortality due to viral GE is low in developed countries but viral GE still accounts for a
considerable morbidity and burden of disease (2).

In the temperate climate zone of the Northern hemisphere, norovirus GII and rotavirus
epidemics peak during the winter season. The onset of the norovirus GII seasonal epidemics
may differ several weeks or even months from year to year, while rotavirus epidemics most
often peak during late winter or early spring (3).

Even though thoroughly studied, the mechanisms behind seasonality of viral GE infections
remain unknown to a large extent. Outdoor climate factors, however, seem to be of
importance to seasonality of viral respiratory infections as well as viral GE infections (3-5).

Previous studies have indicated that low outdoor temperature is associated with annual
norovirus and rotavirus epidemics in temperate climates (3, 5-7). There are, however,
contradicting studies from temperate climates in the southern hemisphere where norovirus
epidemics occur during the warmer months (8, 9). As temperature alone cannot explain the
impact of outdoor climate factors on viral transmission there has been a shift of focus towards
the role of humidity. Low relative humidity (RH; the water content in a gas, relative to the
maximum capacity of water vapor that gas can hold, at a given temperature) has been
associated with increased transmission and viability of both norovirus and rotavirus
(3, 10, 11). In a temperate climate, however, outdoor RH is peaking during winter. Absolute
humidity (AH; the amount of water per volume unit of air (g/m^3), irrespective of
temperature), may be a better predictor of the seasonal variation of influenza and norovirus
detection rate rather than RH (11-13).

The aim of the study was to study the impact of outdoor climate factors on the detection rates
of GE viruses in a large clinical data set, and the association between these factors and the
onset of annual norovirus epidemics in particular.

**Study design**

***Study design and detection of GE agents***

Sahlgrenska University Hospital is the only hospital serving the city of Gothenburg with a
population of approximately 680 000 inhabitants. All clinical samples referred to the hospital
during the seven-year study period (from January 2010 through December 2016), for standard
of care testing by routine multiplex PCR for viral agents causing GE, were included
retrospectively. There was no change in standard of care testing over the course of the study
period. All age groups were represented and included in the analysis. Samples were
predominantly from hospitalized patients, but also from outpatient clinics and primary care
settings to a minor extent. No clinical patient data were available.

Samples were either stool or vomit samples, or rectal swab samples collected with flocked
swabs. As previously described, a multiplex PCR panel was used for detection of six viral
gastrointestinal pathogens (adenovirus, astrovirus, norovirus GI and GII, rotavirus and
sapovirus)(14).

***Meteorological data***

Data on average weekly outdoor temperature (degrees Celsius), vapor pressure (VP; Pa),
absolute humidity (AH; $\text{gH}_2\text{O}/\text{m}^3$), relative humidity (RH; %), wind speed (m/s) and
precipitation (mm) for the study period were obtained from the Swedish Meteorological and
Hydrological Institute's local weather station in Gothenburg (5 m above sea level, situated at
Latitude: 57.7157N, Longitude: 11.9925E).

Detection rates (weekly number of positive samples) of viral agents included in the standard
of care PCR panel and the weekly means of included meteorological parameters were used to
analyze seasonality and correlations. A total of 365 paired, consecutive weekly observations
were included.

Under normal conditions, outside temperature (T), absolute humidity (ρ_v) and vapor pressure
(e) are strongly associated in temperate climates. Absolute humidity is related to vapor
pressure via the ideal gas law for the moist portion of the air:

$e = \rho_v R_v T$ (R_v is the specific gas constant for moist air)

AH and VP are virtually interchangeable as they describe the same phenomenon, one being
mass-based (AH) and the other pressure-based (VP) (15). Due to the strong correspondence of
these three independent variables, potentially undermining the statistical power of the
analysis, only AH was included in the multivariate model.

Via the ideal gas law, AH can also be calculated as a function of T and RH:

$$\rho_v = \frac{6.112 \times e^{\left[\frac{17.67 \times T}{T+243.5}\right]} \times RH \times 2.1674}{(273.15+T)}$$

In order to study weather conditions during the incline, peak and decline of seasonal
epidemics of norovirus GII and rotavirus, subsets of weekly data were extracted. To analyze
these subsets in further detail, positivity rate (the number of positive samples divided by the
total number of samples), rather than detection rate (the number of positive samples), was
used. The viral incline period was defined as the last week with no positive samples to the
117 week of peak positivity rate. The decline period was defined as the week of peak positivity
rate to the first week with no positive samples.

Delayed climate effects on GE virus positivity rates were investigated by introducing a lag-
model in which data from outdoor climate factors were lagged week by week to identify the
optimum threshold.

*Statistical methods*

We used linear regression analysis for univariate comparison of weekly detection rates of
each virus and the meteorological parameters across all seasons. Multiple linear regression
analysis was performed with detection rates of each agent as dependent variables and climate

factors as independent variables. Adjusted R^2 was used to evaluate correlation. Delayed
weather effects on virus positivity rates were analyzed using a lag-model. Optimum
thresholds were identified by repeated regressions, lagging climate factors one week at a time,
until the model's greatest adjusted R^2 was determined. For example, the effect of 'lag one'
represented the effect of the climate factor measured one week prior, on the current week's
positivity rate. Analysis of variance was used to assess significance of each multiple
regression model and a p-value of <0.05 was considered a sufficient model. The statistical
analysis was made using the SPSS software package version 27.0.0.0 (IBM, Armonk, NY).

**Results**

A total of 24 630 samples were analyzed, of which 7 337 (30%) were positive for at least one
viral GE agent. Norovirus genogroup II (GII) was the most common agent detected,
accounting for 4 447 (61%) of the positive samples.

*Seasonality of GE epidemics*

Norovirus GII, sapovirus, rotavirus and astrovirus followed a distinct seasonal pattern with a
peak during the winter season, while norovirus genogroup I (GI) and adenovirus were
detected across all seasons (Figure 1).

Weekly AH, RH, VP, precipitation and temperature all followed seasonal patterns ($p < 0.001$),
while wind speed did not ($p = 0.7$). The seasonal patterns of absolute and relative humidity
are displayed in relation to norovirus GII and rotavirus in figures 2 and 3, respectively.

Univariate analyses comparing weekly detection rate of each virus with meteorological
factors showed that low temperature, low AH and low VP correlated with detection of all six
viruses (Supplementary table I). In addition, a significant correlation was seen between low

relative humidity and rotavirus detection, with borderline significant correlations for
norovirus GII and astrovirus (Supplementary table I).

Results from the linear multiple regression analysis are shown in table 1. Low AH and low
temperature were significantly associated with detection of all agents in the panel.

*Impact of humidity on beginning and ending of noro- and rotavirus epidemics*

Every season, the start of the norovirus epidemic coincided with a sharp drop in AH and the
decline with a distinct rise in AH. To study these correlations in further detail, we analyzed
weekly norovirus positivity rates during the incline (figure 4) and decline period (figure 5) of
every season.

The strongest correlations between outdoor climate factors and weekly positivity rates were
seen when lagging weather factors several weeks. For norovirus GII univariate regression
models showed the lowest variance (greatest adjusted R^2) to AH and temperature when
lagging weather data four weeks, corresponding to a four-week period between a drop in
outside AH and temperature before reaching peak positivity rate of the seasonal norovirus
epidemic. When assessing rotavirus using the same lagging strategy the strongest correlation
was seen at 6- and 10-week lags for AH and temperature respectively (Supplementary figure
I).

We observed longer lag time optimums when comparing norovirus and rotavirus to RH, 12
and 14 weeks respectively. AH had greater lag time optimum correlations with norovirus GII
and rotavirus than RH (adjusted R^2 : 0.47 and 0.37 versus 0.24 and 0.20, respectively; p-
values: 3×10^{-48} and 8×10^{-34} versus 4×10^{-21} and 5×10^{-18} , respectively).

**Discussion**

We show that a drop in outside absolute humidity and temperature precede the rise of
seasonal norovirus and rotavirus positivity rates, while a rise in AH and temperature precedes
falling norovirus and rotavirus positivity rates.

Detection rates of viral GE agents showed a stronger correlation with AH compared to RH,
which is consistent with findings in viral respiratory infections such as influenza (12).

Previous studies have mostly focused on temperature, precipitation and RH to describe the
relationship between outdoor climate factors and incidence of viral respiratory and GE
infections, but recent publications indicate that outdoor AH may be more relevant than
outdoor RH to predict a seasonal rise in influenza (12, 13) and norovirus (11) infections. In
agreement with our findings, Atchison et al have shown a strong inverse correlation between
temperature and rotavirus detection, and a weaker inverse correlation between RH and
rotavirus detection (16). Though, in contrast to our findings, they found no associations
between AH and rotavirus detection.

Several mechanisms can contribute to the increased spread of norovirus in a setting with low
AH as seen in our study. Colas de la Noue et al have shown that low AH favored murine
norovirus survival and infectivity (and norovirus VLP stability) in the environment (11).
Evaporation of aerosolized droplets has been suggested as a possible mechanism (10, 17, 18),
since small droplets, after vomiting, may remain aerosolized for a longer time and travel
further distance when AH is low. The combination of larger contaminated areas and increased
viral survival and stability on surfaces could affect transmission via the fecal-oral route and
contribute to the annual increase in norovirus infections during wintertime in a temperate
climate.

Other factors, such as antigenic variation (3), declining herd immunity (3, 5), long term
shedding (19) and social behavior such as indoor crowding, may also contribute to the
seasonal variation of norovirus, as well as other viral pathogens.

We found that norovirus GII, rotavirus, sapovirus and astrovirus all showed distinct seasonal
patterns of detection with peaks during the winter or early spring season, whereas norovirus
GI and adenovirus were more evenly distributed across all seasons. For adenovirus, the lack
of seasonality is in line with previous studies (20, 21). Norovirus GI is associated with water-
and foodborne outbreaks and has less potential for person-to-person spread, which could
explain why no distinct seasonality was observed (22).

We also found a correlation between RH and rotavirus positivity rates. However, this
correlation, previously described in other reports (10, 23, 24), is more likely to be explained
by a drop in temperature and AH. The RH minimum and rotavirus peak positivity rate both
lag several weeks compared to temperature and AH minimums, generating a possible
relationship between RH and rotavirus positivity rate. However, as exemplified by the 2011,
2013 and 2016 rotavirus seasons, the rise in rotavirus positivity rates occasionally preceded
the drop in RH, which is inconsistent with a causal effect of RH on transmission.

Nevertheless, AH remained the strongest predictor among the outdoor weather factors.

As transmission of viruses often occurs indoors, outdoor climate factors cannot fully explain
seasonal variations in detection rates of viral infections. It has previously been suggested that
mass-based outdoor humidity, such as AH and specific humidity, serves as the best proxy for
indoor climate conditions (25). Unlike RH or temperature, AH shows a high correlation with
indoor and outdoor settings, indicating that it might be more useful as a predictor of viral
transmission, especially since it is not a temperature-dependent ratio (12, 25-27). It has also
been suggested that AH should be preferred over RH in medical research (15).

AH correlates well with virus transmission, although the mechanisms are difficult to explain.

In an attempt to explain the relationship between virus viability and climate factors it may be
preferable to study AH as a function of RH and temperature, using the ideal gas law. RH
affects droplet size and eventual rupture, while increased temperature inactivates exposed

nucleic acids and proteins, which may explain the combined effects of AH on virus viability
(28).

The strongest correlations between outdoor climate factors and viral positivity rate were seen
when lagging outdoor climate factors several weeks, which is in line with previous reports
(16, 23). The lag times were generally longer in our study, which could be explained by
differences between temperate and subtropical settings, the impact of other outdoor and
indoor climate factors or population density (29, 30). In our study, norovirus reached peak
positivity rate in approximately half the time of rotavirus, when counting from the largest
drop in AH (and temperature). The observation that rotavirus epidemics develop more slowly,
compared to norovirus epidemics, under the influence of outdoor climate factors is in
agreement with previous observations (23). Numerous factors likely contribute to differing
lag times for each virus. Norovirus GII has a shorter incubation period and a larger proportion
of asymptomatic carriers than rotavirus (31), and the forceful vomiting typical for norovirus
disease contributes to efficient spread. Furthermore, adolescents and adults are susceptible to
norovirus since immunity is genotype-, or even strain-specific and not as long-lasting as
immunity to rotavirus, generating a much larger susceptible population (32).

This study has several limitations. Firstly, we have not been able to adjust for confounders
such as herd immunity, viral evolution and emergence of new viral strains, or behavioral
factors such as school holidays and indoor crowding, which could affect the results.

Secondly, we only included a limited number of outdoor climate factors. Other factors, such
as UV radiation, could also affect viral transmission (33).

Finally, this study was performed using data from one meteorological station and a single
clinical center and our observations may not be applicable to other regions than our
geographical setting.

Additional studies are needed to illuminate the impact of humidity and other climate factors
on GE virus transmission. As transmission mainly occurs indoors, it would be of interest to
study indoor climate factors in relation to incidence of viral gastroenteritis.

In conclusion, we found a strong correlation between low absolute humidity and norovirus
GII and rotavirus positivity rates. Absolute humidity was a better predictor of positivity rates
of norovirus GII and rotavirus than relative humidity. A sharp drop in absolute humidity
preceded the increase in norovirus GII positivity rate each season and peak positivity rate was
reached four weeks after the drop in absolute humidity. Absolute humidity may be used as a
predictor of viral gastroenteritis seasonality in a temperate climate.

**Funding**

This study was supported by grants from the Gothenburg Medical Society (grant number
GLS-935274)

**Competing interests**

None declared

**Ethical approval**

Not required. All samples were collected for clinical purposes. Data was deidentified prior to
analysis and no personal information was available.

References

1. Kassebaum NJ, Bertozzi-Villa A, Coggeshall MS, Shackelford KA, Steiner C, Heuton KR, et al. Global, regional, and national levels and causes of maternal mortality during 1990-2013: a systematic analysis for the Global Burden of Disease Study 2013. *Lancet*. 2014;384(9947):980-1004.
2. Siebenga JJ, Vennema H, Zheng DP, Vinje J, Lee BE, Pang XL, et al. Norovirus illness is a global problem: emergence and spread of norovirus GII.4 variants, 2001-2007. *J Infect Dis*. 2009;200(5):802-12.
3. Lopman B, Armstrong B, Atchison C, Gray JJ. Host, weather and virological factors drive norovirus epidemiology: time-series analysis of laboratory surveillance data in England and Wales. *PLoS One*. 2009;4(8):e6671.
4. Sundell N, Andersson LM, Brittain-Long R, Lindh M, Westin J. A four year seasonal survey of the relationship between outdoor climate and epidemiology of viral respiratory tract infections in a temperate climate. *J Clin Virol*. 2016;84:59-63.
5. van Gaalen RD, van de Kasstele J, Hahné SJM, Bruijning-Verhagen P, Wallinga J. Determinants of Rotavirus Transmission: A Lag Nonlinear Time Series Analysis. *Epidemiology*. 2017;28(4):503-13.
6. Hervás D, Hervás-Masip J, Rosell A, Mena A, Pérez JL, Hervás JA. Are hospitalizations for rotavirus gastroenteritis associated with meteorologic factors? *Eur J Clin Microbiol Infect Dis*. 2014;33(9):1547-53.
7. Mounts AW, Ando T, Koopmans M, Bresee JS, Noel J, Glass RI. Cold weather seasonality of gastroenteritis associated with Norwalk-like viruses. *J Infect Dis*. 2000;181 Suppl 2:S284-7.
8. Bruggink LD, Marshall JA. Norovirus epidemics are linked to two distinct sets of controlling factors. *Int J Infect Dis*. 2009;13(3):e125-6.
9. Marshall JA, Dimitriadis A, Wright PJ. Molecular and epidemiological features of norovirus-associated gastroenteritis outbreaks in Victoria, Australia in 2001. *J Med Virol*. 2005;75(2):321-31.
10. Sattar SA, Ijaz MK, Johnson-Lussenburg CM, Springthorpe VS. Effect of relative humidity on the airborne survival of rotavirus SA11. *Appl Environ Microbiol*. 1984;47(4):879-81.
11. Colas de la Noue A, Estienney M, Aho S, Perrier-Cornet JM, de Rougemont A, Pothier P, et al. Absolute Humidity Influences the Seasonal Persistence and Infectivity of Human Norovirus. *Appl Environ Microbiol*. 2014;80(23):7196-205.
12. Shaman J, Kohn M. Absolute humidity modulates influenza survival, transmission, and seasonality. *Proc Natl Acad Sci U S A*. 2009;106(9):3243-8.
13. Shaman J, Kandula S, Yang W, Karspeck A. The use of ambient humidity conditions to improve influenza forecast. *PLoS Comput Biol*. 2017;13(11):e1005844.
14. Gustavsson L, Westin J, Andersson LM, Lindh M. Rectal swabs can be used for diagnosis of viral gastroenteritis with a multiple real-time PCR assay. *J Clin Virol*. 2011;51(4):279-82.
15. Davis RE, McGregor GR, Enfield KB. Humidity: A review and primer on atmospheric moisture and human health. *Environ Res*. 2016;144(Pt A):106-16.
16. Atchison CJ, Tam CC, Hajat S, van Pelt W, Cowden JM, Lopman BA. Temperature-dependent transmission of rotavirus in Great Britain and The Netherlands. *Proc Biol Sci*. 2010;277(1683):933-42.

17. Jones RM, Brosseau LM. Aerosol transmission of infectious disease. *J Occup Environ Med.* 2015;57(5):501-8.
18. Alsveld M, Widell A, Dahlin H, Karlson S, Medstrand P, Londahl J. Aerosolization and recovery of viable murine norovirus in an experimental setup. *Sci Rep.* 2020;10(1):15941.
19. Marshall JA, Bruggink LD. The dynamics of norovirus outbreak epidemics: recent insights. *Int J Environ Res Public Health.* 2011;8(4):1141-9.
20. Feeney SA, Armstrong VJ, Mitchell SJ, Crawford L, McCaughey C, Coyle PV. Development and clinical validation of multiplex TaqMan(R) assays for rapid diagnosis of viral gastroenteritis. *J Med Virol.* 2011;83(9):1650-6.
21. Pang XL, Honma S, Nakata S, Vesikari T. Human caliciviruses in acute gastroenteritis of young children in the community. *J Infect Dis.* 2000;181 Suppl 2:S288-94.
22. Lysen M, Thorhagen M, Brytting M, Hjertqvist M, Andersson Y, Hedlund KO. Genetic diversity among food-borne and waterborne norovirus strains causing outbreaks in Sweden. *J Clin Microbiol.* 2009;47(8):2411-8.
23. Wang P, Goggins WB, Chan EYY. A time-series study of the association of rainfall, relative humidity and ambient temperature with hospitalizations for rotavirus and norovirus infection among children in Hong Kong. *Sci Total Environ.* 2018;643:414-22.
24. Levy K, Hubbard AE, Eisenberg JN. Seasonality of rotavirus disease in the tropics: a systematic review and meta-analysis. *Int J Epidemiol.* 2009;38(6):1487-96.
25. Nguyen JL, Dockery DW. Daily indoor-to-outdoor temperature and humidity relationships: a sample across seasons and diverse climatic regions. *Int J Biometeorol.* 2016;60(2):221-9.
26. Quinn A, Shaman J. Indoor temperature and humidity in New York City apartments during winter. *Sci Total Environ.* 2017;583:29-35.
27. Nguyen JL, Schwartz J, Dockery DW. The relationship between indoor and outdoor temperature, apparent temperature, relative humidity, and absolute humidity. *Indoor Air.* 2014;24(1):103-12.
28. Marr LC, Tang JW, Van Mullekom J, Lakdawala SS. Mechanistic insights into the effect of humidity on airborne influenza virus survival, transmission and incidence. *J R Soc Interface.* 2019;16(150):20180298.
29. Kiulia NM, Hofstra N, Vermeulen LC, Obara MA, Medema G, Rose JB. Global occurrence and emission of rotaviruses to surface waters. *Pathogens.* 2015;4(2):229-55.
30. Boxman IL, Verhoef L, Dijkman R, Hagele G, Te Loeke NA, Koopmans M. Year-round prevalence of norovirus in the environment of catering companies without a recently reported outbreak of gastroenteritis. *Appl Environ Microbiol.* 2011;77(9):2968-74.
31. Quee FA, de Hoog MLA, Schuurman R, Bruijning-Verhagen P. Community burden and transmission of acute gastroenteritis caused by norovirus and rotavirus in the Netherlands (RotaFam): a prospective household-based cohort study. *Lancet Infect Dis.* 2020;20(5):598-606.
32. Lopman BA, Steele D, Kirkwood CD, Parashar UD. The Vast and Varied Global Burden of Norovirus: Prospects for Prevention and Control. *PLoS Med.* 2016;13(4):e1001999.
33. Park D, Shahbaz HM, Kim SH, Lee M, Lee W, Oh JW, et al. Inactivation efficiency and mechanism of UV-TiO₂ photocatalysis against murine norovirus using a solidified agar matrix. *Int J Food Microbiol.* 2016;238:256-64.

Response to reviewers

Spectrum02433-22 Outdoor absolute humidity predicts seasonal variation of norovirus GII infection.

Reviewer #1:

We welcome all suggestions and have edited the paper accordingly.

Line 11: reword 'outdoor climate factors' with 'weather conditions'.

Answer: The wording has been changed as suggested by reviewer #1.

Line 12: replace 'between weather conditions and' with 'with'.

Answer: The wording has been changed as suggested by reviewer #1.

Line 14: delete 'agents causing viral' and add 'viruses' after 'gastroenteritis'.

Answer: The wording has been changed as suggested by reviewer #1.

Line 20: replace 'agents with 'viruses'.

Answer: The wording has been changed as suggested by reviewer #1.

Line 24: AH: spell out in full the first time used.

Answer: The wording has been changed as suggested by reviewer #1.

Line 35: would replace 'impact', here and in line 70 with 'association'

Answer: The wording has been changed as suggested by reviewer #1.

Line 86: is the assay specific for enteric (group F) adenoviruses or for all adenoviruses?

Answer: The assay is not specific for enteric (group F) adenovirus, it detects all adenoviruses. Group F (40/41) analysis is occasionally performed in specific cases.

Reviewer #2:

1. *The use of a linear model (univariate or multiple) is appropriate for an outcome variable that is a proportion (positivity rate) or count (number of cases), and generalized linear model should be used instead.*

Answer: We agree that linear regression might not be suitable when the type outcome is a count or a proportion, since the distribution of count or proportion data can be skewed. In the present study, we have checked the distribution of the residuals from the linear regression analysis. The distribution is close to normal and symmetric in most cases. Therefore, we decided, in collaboration with an experienced medical statistician, that it was appropriate to use linear regression in our analysis.

2. *Further, with the use of time-series data the impact of autocorrelation and for meteorological data, seasonality, need to be explored and likely incorporated into the model.*

Answer: We agree. Autocorrelation and seasonality may skew the data, which is better accounted for in a mixed model. We therefore checked our data by doing identical linear regression models on randomly selected subsets of measuring points (weeks) across all six seasons. The models created by these random measuring points gave the same directions

in coefficients, indicating a low impact of autocorrection and seasonality on the overall conclusions.

3. *There does not appear to be a clear strategy for determining which meteorological variables and lag time lengths for each variable should be included in each of the multivariable models.*

Answer: We agree that the presentation of variable selection was incomplete. This has been clarified in the Study design – Meteorological data section (Lines 109-116).

Determination of lag time lengths is described in detail in the Statistical methods section (Lines 131-136).

October 17, 2022

Dr. Thomas Beck-Friis
Biomedicine, Sahlgrenska Academy, University of Gothenburg
Infectious Diseases
Diagnosvägen 21
Gothenburg 41650
Sweden

Re: Spectrum02433-22R1 (Outdoor absolute humidity predicts seasonal variation of norovirus GII infection.)

Dear Dr. Thomas Beck-Friis:

Thank you for submitting your revised manuscript to Microbiology Spectrum. One of the reviewers, which is an expert on virus epidemiology, is still concerned about the model used to support your conclusions. I am willing to consider a second round of revisions if you perform the analyses with the GLMM model suggested by the reviewer. Copies of the reviewers' comments are below for your consideration and review.

Link Not Available

Sincerely,

Gabriel Parra

Journals Department
Reviewer comments:

Reviewer #2 (Comments for the Author):

Thank you for your justifications of your statistical analysis. I still feel that linear regression is not appropriate based on your outcome measure, detection rates, since it is not a continuous variable. I would strongly consider using a generalized linear mixed model, so you can accurately model your outcome and account for the temporal correlation you have with your repeated measurements and seasonality. Your data is not independent and needs to be accounted for in the model.

Staff Comments:

Preparing Revision Guidelines

Please return the manuscript within 60 days; if you cannot complete the modification within this time period, please contact me. If you do not wish to modify the manuscript and prefer to submit it to another journal, please notify me of your decision immediately so that the manuscript may be formally withdrawn from consideration by Microbiology Spectrum.

**Outdoor absolute humidity predicts seasonal variation of norovirus GII infection.**

Thomas Beck-Friis^{1,2*}, Nicklas Sundell^{1,2}, Lars Gustavsson^{1,2}, Magnus Lindh^{1,3}, Johan

Westin^{1,2}, Lars-Magnus Andersson^{1,2}

¹*Department of Infectious Diseases, Sahlgrenska University Hospital, Region Västra Götaland, Gothenburg,*
*Sweden*

²*Department of Infectious Diseases, Institute of Biomedicine, University of Gothenburg, Gothenburg, Sweden*

³*Department of Microbiology, Institute of Biomedicine, University of Gothenburg, Gothenburg, Sweden*

**Abstract**

*Background:* Seasonal variation of viral gastroenteritis is related to outdoor climate factors,
but the relationship between weather conditions and the incidence of viral gastroenteritis (GE)
is not fully understood.

*Objectives:* To examine the impact of outdoor climate factors on seasonal
variation in detection rates of agents causing viral gastroenteritis, with emphasis on norovirus.

*Study design:* Weekly detection rates of norovirus genogroup I (GI) and II (GII), rotavirus,
adenovirus, astrovirus and sapovirus detection were analyzed in relation to average weekly
means of included meteorological parameters. Associations between rates of PCR detection of
the viral GE pathogens and climate factors were investigated with linear regression models.

*Results:* Low absolute humidity and low temperature were correlated with increased detection
of all agents in the panel, (adjusted R²): adenovirus (0.036; p=0.0002), astrovirus (0.13; p=2 x
10⁻¹²), rotavirus (0.14; p=4 x 10⁻¹³), norovirus GI (0.13; p=4 x 10⁻¹²), norovirus GII (0.36; p=2
23 x 10⁻³⁴), and sapovirus (0.12; p=4 x 10⁻¹¹). Detection of norovirus GII, rotavirus, astrovirus
and sapovirus followed seasonal patterns, while adenovirus and norovirus GI detection did
not. In each investigated season, a sharp drop in AH preceded the increase in norovirus

positivity rate and peak positivity rate was reached four weeks after the drop in absolute
humidity.

*Conclusions:* We found a strong correlation between low AH and norovirus GII positivity
rate. Absolute humidity was a better predictor of positivity rates of norovirus GII and
rotavirus than relative humidity.

**Importance:**

Viral gastroenteritis causes considerable morbidity, especially in vulnerable groups such as
the elderly and chronically ill. Predicting the beginning of seasonal epidemics is important for
the health care system to withstand the increasing demands.

In this paper we studied the impact of outdoor climate factors on the detection rates of
gastrointestinal viruses and the association between these factors and the onset of annual
norovirus epidemics.

A sharp fall in absolute humidity preceded the increase in diagnosed norovirus cases by
approximately four weeks. These findings contribute to the understanding of norovirus
epidemiology and allow health care services to install timely preventive measures and can
help the public avoid transmission.

**Keywords (MeSH):** Norovirus, rotavirus, weather / epidemiology, humidity, gastroenteritis /
epidemiology

**Background**

Acute viral gastroenteritis is a major cause of morbidity and mortality worldwide (1).

Mortality due to viral GE is low in developed countries but viral GE still accounts for a
considerable morbidity and burden of disease (2).

In the temperate climate zone of the Northern hemisphere, norovirus GII and rotavirus
epidemics peak during the winter season. The onset of the norovirus GII seasonal epidemics
may differ several weeks or even months from year to year, while rotavirus epidemics most
often peak during late winter or early spring (3).

Even though thoroughly studied, the mechanisms behind seasonality of viral GE infections
remain unknown to a large extent. Outdoor climate factors, however, seem to be of
importance to seasonality of viral respiratory infections as well as viral GE infections (3-5).

Previous studies have indicated that low outdoor temperature is associated with annual
norovirus and rotavirus epidemics in temperate climates (3, 5-7). There are, however,
contradicting studies from temperate climates in the southern hemisphere where norovirus
epidemics occur during the warmer months (8, 9). As temperature alone cannot explain the
impact of outdoor climate factors on viral transmission there has been a shift of focus towards
the role of humidity. Low relative humidity (RH; the water content in a gas, relative to the
maximum capacity of water vapor that gas can hold, at a given temperature) has been
associated with increased transmission and viability of both norovirus and rotavirus
(3, 10, 11). In a temperate climate, however, outdoor RH is peaking during winter. Absolute
humidity (AH; the amount of water per volume unit of air (g/m^3), irrespective of
temperature), may be a better predictor of the seasonal variation of influenza and norovirus
detection rate rather than RH (11-13).

The aim of the study was to study the impact of outdoor climate factors on the detection rates
of GE viruses in a large clinical data set, and the association between these factors and the
onset of annual norovirus epidemics in particular.

**Study design**

***Study design and detection of GE agents***

Sahlgrenska University Hospital is the only hospital serving the city of Gothenburg with a
population of approximately 680 000 inhabitants. All clinical samples referred to the hospital
during the seven-year study period (from January 2010 through December 2016), for standard
of care testing by routine multiplex PCR for viral agents causing GE, were included
retrospectively. There was no change in standard of care testing over the course of the study
period. All age groups were represented and included in the analysis. Samples were
predominantly from hospitalized patients, but also from outpatient clinics and primary care
settings to a minor extent. No clinical patient data were available.

Samples were either stool or vomit samples, or rectal swab samples collected with flocked
swabs. As previously described, a multiplex PCR panel was used for detection of six viral
gastrointestinal pathogens (adenovirus, astrovirus, norovirus GI and GII, rotavirus and
sapovirus)(14).

***Meteorological data***

Data on average weekly outdoor temperature (degrees Celsius), vapor pressure (VP; Pa),
absolute humidity (AH; gH₂O/m³), relative humidity (RH; %), wind speed (m/s) and
precipitation (mm) for the study period were obtained from the Swedish Meteorological and
Hydrological Institute's local weather station in Gothenburg (5 m above sea level, situated at
Latitude: 57.7157N, Longitude: 11.9925E).

Detection rates (weekly number of positive samples) of viral agents included in the standard
of care PCR panel and the weekly means of included meteorological parameters were used to
analyze seasonality and correlations. A total of 365 paired, consecutive weekly observations
were included.

Under normal conditions, outside temperature (T), absolute humidity (ρ_v) and vapor pressure
(e) are strongly associated in temperate climates. Absolute humidity is related to vapor
pressure via the ideal gas law for the moist portion of the air:
$e = \rho_v R_v T$ (R_v is the specific gas constant for moist air)
AH and VP are virtually interchangeable as they describe the same phenomenon, one being
mass-based (AH) and the other pressure-based (VP) (15). Due to the strong correspondence of
these three independent variables, potentially undermining the statistical power of the
analysis, only AH was included in the multivariate model.

Via the ideal gas law, AH can also be calculated as a function of T and RH:

$$109 \rho_v = \frac{6.112 \times e^{[(17.67 \times T)/(T+243.5)]} \times RH \times 2.1674}{110 (273.15+T)}$$

In order to study weather conditions during the incline, peak and decline of seasonal
epidemics of norovirus GII and rotavirus, subsets of weekly data were extracted. To analyze
these subsets in further detail, positivity rate (the number of positive samples divided by the
total number of samples), rather than detection rate (the number of positive samples), was
used. The viral incline period was defined as the last week with no positive samples to the
117 week of peak positivity rate. The decline period was defined as the week of peak positivity
rate to the first week with no positive samples.

Delayed climate effects on GE virus positivity rates were investigated by introducing a lag-
model in which data from outdoor climate factors were lagged week by week to identify the
optimum threshold.

*Statistical methods*

We used linear regression analysis for univariate comparison of weekly detection rates of
each virus and the meteorological parameters across all seasons. Multiple linear regression
analysis was performed with detection rates of each agent as dependent variables and climate

factors as independent variables. Adjusted R^2 was used to evaluate correlation. Delayed
weather effects on virus positivity rates were analyzed using a lag-model. Optimum
thresholds were identified by repeated regressions, lagging climate factors one week at a time,
until the model's greatest adjusted R^2 was determined. For example, the effect of 'lag one'
represented the effect of the climate factor measured one week prior, on the current week's
positivity rate. Analysis of variance was used to assess significance of each multiple
regression model and a p-value of <0.05 was considered a sufficient model. The statistical
analysis was made using the SPSS software package version 27.0.0.0 (IBM, Armonk, NY).

**Results**

A total of 24 630 samples were analyzed, of which 7 337 (30%) were positive for at least one
viral GE agent. Norovirus genogroup II (GII) was the most common agent detected,
accounting for 4 447 (61%) of the positive samples.

*Seasonality of GE epidemics*

Norovirus GII, sapovirus, rotavirus and astrovirus followed a distinct seasonal pattern with a
peak during the winter season, while norovirus genogroup I (GI) and adenovirus were
detected across all seasons (Figure 1).

Weekly AH, RH, VP, precipitation and temperature all followed seasonal patterns ($p < 0.001$),
while wind speed did not ($p = 0.7$). The seasonal patterns of absolute and relative humidity
are displayed in relation to norovirus GII and rotavirus in figures 2 and 3, respectively.

Univariate analyses comparing weekly detection rate of each virus with meteorological
factors showed that low temperature, low AH and low VP correlated with detection of all six
viruses (Supplementary table I). In addition, a significant correlation was seen between low

relative humidity and rotavirus detection, with borderline significant correlations for
norovirus GII and astrovirus (Supplementary table I).

Results from the linear multiple regression analysis are shown in table 1. Low AH and low
temperature were significantly associated with detection of all agents in the panel.

*Impact of humidity on beginning and ending of noro- and rotavirus epidemics*

Every season, the start of the norovirus epidemic coincided with a sharp drop in AH and the
decline with a distinct rise in AH. To study these correlations in further detail, we analyzed
weekly norovirus positivity rates during the incline (figure 4) and decline period (figure 5) of
every season.

The strongest correlations between outdoor climate factors and weekly positivity rates were
seen when lagging weather factors several weeks. For norovirus GII univariate regression
models showed the lowest variance (greatest adjusted R^2) to AH and temperature when
lagging weather data four weeks, corresponding to a four-week period between a drop in
outside AH and temperature before reaching peak positivity rate of the seasonal norovirus
epidemic. When assessing rotavirus using the same lagging strategy the strongest correlation
was seen at 6- and 10-week lags for AH and temperature respectively (Supplementary figure
I).

We observed longer lag time optimums when comparing norovirus and rotavirus to RH, 12
and 14 weeks respectively. AH had greater lag time optimum correlations with norovirus GII
and rotavirus than RH (adjusted R^2 : 0.47 and 0.37 versus 0.24 and 0.20, respectively; p-
values: 3×10^{-48} and 8×10^{-34} versus 4×10^{-21} and 5×10^{-18} , respectively).

**Discussion**

We show that a drop in outside absolute humidity and temperature precede the rise of
seasonal norovirus and rotavirus positivity rates, while a rise in AH and temperature precedes
falling norovirus and rotavirus positivity rates.

Detection rates of viral GE agents showed a stronger correlation with AH compared to RH,
which is consistent with findings in viral respiratory infections such as influenza (12).

Previous studies have mostly focused on temperature, precipitation and RH to describe the
relationship between outdoor climate factors and incidence of viral respiratory and GE
infections, but recent publications indicate that outdoor AH may be more relevant than
outdoor RH to predict a seasonal rise in influenza (12, 13) and norovirus (11) infections. In
agreement with our findings, Atchison et al have shown a strong inverse correlation between
temperature and rotavirus detection, and a weaker inverse correlation between RH and
rotavirus detection (16). Though, in contrast to our findings, they found no associations
between AH and rotavirus detection.

Several mechanisms can contribute to the increased spread of norovirus in a setting with low
AH as seen in our study. Colas de la Noue et al have shown that low AH favored murine
norovirus survival and infectivity (and norovirus VLP stability) in the environment (11).
Evaporation of aerosolized droplets has been suggested as a possible mechanism (10, 17, 18),
since small droplets, after vomiting, may remain aerosolized for a longer time and travel
further distance when AH is low. The combination of larger contaminated areas and increased
viral survival and stability on surfaces could affect transmission via the fecal-oral route and
contribute to the annual increase in norovirus infections during wintertime in a temperate
climate.

Other factors, such as antigenic variation (3), declining herd immunity (3, 5), long term
shedding (19) and social behavior such as indoor crowding, may also contribute to the
seasonal variation of norovirus, as well as other viral pathogens.

We found that norovirus GII, rotavirus, sapovirus and astrovirus all showed distinct seasonal
patterns of detection with peaks during the winter or early spring season, whereas norovirus
GI and adenovirus were more evenly distributed across all seasons. For adenovirus, the lack
of seasonality is in line with previous studies (20, 21). Norovirus GI is associated with water-
and foodborne outbreaks and has less potential for person-to-person spread, which could
explain why no distinct seasonality was observed (22).

We also found a correlation between RH and rotavirus positivity rates. However, this
correlation, previously described in other reports (10, 23, 24), is more likely to be explained
by a drop in temperature and AH. The RH minimum and rotavirus peak positivity rate both
lag several weeks compared to temperature and AH minimums, generating a possible
relationship between RH and rotavirus positivity rate. However, as exemplified by the 2011,
2013 and 2016 rotavirus seasons, the rise in rotavirus positivity rates occasionally preceded
the drop in RH, which is inconsistent with a causal effect of RH on transmission.

Nevertheless, AH remained the strongest predictor among the outdoor weather factors.

As transmission of viruses often occurs indoors, outdoor climate factors cannot fully explain
seasonal variations in detection rates of viral infections. It has previously been suggested that
mass-based outdoor humidity, such as AH and specific humidity, serves as the best proxy for
indoor climate conditions (25). Unlike RH or temperature, AH shows a high correlation with
indoor and outdoor settings, indicating that it might be more useful as a predictor of viral
transmission, especially since it is not a temperature-dependent ratio (12, 25-27). It has also
been suggested that AH should be preferred over RH in medical research (15).

AH correlates well with virus transmission, although the mechanisms are difficult to explain.

In an attempt to explain the relationship between virus viability and climate factors it may be
preferable to study AH as a function of RH and temperature, using the ideal gas law. RH
affects droplet size and eventual rupture, while increased temperature inactivates exposed

nucleic acids and proteins, which may explain the combined effects of AH on virus viability
(28).

The strongest correlations between outdoor climate factors and viral positivity rate were seen
when lagging outdoor climate factors several weeks, which is in line with previous reports
(16, 23). The lag times were generally longer in our study, which could be explained by
differences between temperate and subtropical settings, the impact of other outdoor and
indoor climate factors or population density (29, 30). In our study, norovirus reached peak
positivity rate in approximately half the time of rotavirus, when counting from the largest
drop in AH (and temperature). The observation that rotavirus epidemics develop more slowly,
compared to norovirus epidemics, under the influence of outdoor climate factors is in
agreement with previous observations (23). Numerous factors likely contribute to differing
lag times for each virus. Norovirus GII has a shorter incubation period and a larger proportion
of asymptomatic carriers than rotavirus (31), and the forceful vomiting typical for norovirus
disease contributes to efficient spread. Furthermore, adolescents and adults are susceptible to
norovirus since immunity is genotype-, or even strain-specific and not as long-lasting as
immunity to rotavirus, generating a much larger susceptible population (32).

This study has several limitations. Firstly, we have not been able to adjust for confounders
such as herd immunity, viral evolution and emergence of new viral strains, or behavioral
factors such as school holidays and indoor crowding, which could affect the results.

Secondly, we only included a limited number of outdoor climate factors. Other factors, such
as UV radiation, could also affect viral transmission (33).

Finally, this study was performed using data from one meteorological station and a single
clinical center and our observations may not be applicable to other regions than our
geographical setting.

Additional studies are needed to illuminate the impact of humidity and other climate factors
on GE virus transmission. As transmission mainly occurs indoors, it would be of interest to
study indoor climate factors in relation to incidence of viral gastroenteritis.

In conclusion, we found a strong correlation between low absolute humidity and norovirus
GII and rotavirus positivity rates. Absolute humidity was a better predictor of positivity rates
of norovirus GII and rotavirus than relative humidity. A sharp drop in absolute humidity
preceded the increase in norovirus GII positivity rate each season and peak positivity rate was
reached four weeks after the drop in absolute humidity. Absolute humidity may be used as a
predictor of viral gastroenteritis seasonality in a temperate climate.

**Funding**

This study was supported by grants from the Gothenburg Medical Society (grant number
GLS-935274)

**Competing interests**

None declared

**Ethical approval**

Not required. All samples were collected for clinical purposes. Data was deidentified prior to
analysis and no personal information was available.

References

1. Kassebaum NJ, Bertozzi-Villa A, Coggeshall MS, Shackelford KA, Steiner C, Heuton KR, et al. Global, regional, and national levels and causes of maternal mortality during 1990-2013: a systematic analysis for the Global Burden of Disease Study 2013. *Lancet*. 2014;384(9947):980-1004.
2. Siebenga JJ, Vennema H, Zheng DP, Vinje J, Lee BE, Pang XL, et al. Norovirus illness is a global problem: emergence and spread of norovirus GII.4 variants, 2001-2007. *J Infect Dis*. 2009;200(5):802-12.
3. Lopman B, Armstrong B, Atchison C, Gray JJ. Host, weather and virological factors drive norovirus epidemiology: time-series analysis of laboratory surveillance data in England and Wales. *PLoS One*. 2009;4(8):e6671.
4. Sundell N, Andersson LM, Brittain-Long R, Lindh M, Westin J. A four year seasonal survey of the relationship between outdoor climate and epidemiology of viral respiratory tract infections in a temperate climate. *J Clin Virol*. 2016;84:59-63.
5. van Gaalen RD, van de Kasstele J, Hahné SJM, Bruijning-Verhagen P, Wallinga J. Determinants of Rotavirus Transmission: A Lag Nonlinear Time Series Analysis. *Epidemiology*. 2017;28(4):503-13.
6. Hervás D, Hervás-Masip J, Rosell A, Mena A, Pérez JL, Hervás JA. Are hospitalizations for rotavirus gastroenteritis associated with meteorologic factors? *Eur J Clin Microbiol Infect Dis*. 2014;33(9):1547-53.
7. Mounts AW, Ando T, Koopmans M, Bresee JS, Noel J, Glass RI. Cold weather seasonality of gastroenteritis associated with Norwalk-like viruses. *J Infect Dis*. 2000;181 Suppl 2:S284-7.
8. Bruggink LD, Marshall JA. Norovirus epidemics are linked to two distinct sets of controlling factors. *Int J Infect Dis*. 2009;13(3):e125-6.
9. Marshall JA, Dimitriadis A, Wright PJ. Molecular and epidemiological features of norovirus-associated gastroenteritis outbreaks in Victoria, Australia in 2001. *J Med Virol*. 2005;75(2):321-31.
10. Sattar SA, Ijaz MK, Johnson-Lussenburg CM, Springthorpe VS. Effect of relative humidity on the airborne survival of rotavirus SA11. *Appl Environ Microbiol*. 1984;47(4):879-81.
11. Colas de la Noue A, Estienney M, Aho S, Perrier-Cornet JM, de Rougemont A, Pothier P, et al. Absolute Humidity Influences the Seasonal Persistence and Infectivity of Human Norovirus. *Appl Environ Microbiol*. 2014;80(23):7196-205.
12. Shaman J, Kohn M. Absolute humidity modulates influenza survival, transmission, and seasonality. *Proc Natl Acad Sci U S A*. 2009;106(9):3243-8.
13. Shaman J, Kandula S, Yang W, Karspeck A. The use of ambient humidity conditions to improve influenza forecast. *PLoS Comput Biol*. 2017;13(11):e1005844.
14. Gustavsson L, Westin J, Andersson LM, Lindh M. Rectal swabs can be used for diagnosis of viral gastroenteritis with a multiple real-time PCR assay. *J Clin Virol*. 2011;51(4):279-82.
15. Davis RE, McGregor GR, Enfield KB. Humidity: A review and primer on atmospheric moisture and human health. *Environ Res*. 2016;144(Pt A):106-16.
16. Atchison CJ, Tam CC, Hajat S, van Pelt W, Cowden JM, Lopman BA. Temperature-dependent transmission of rotavirus in Great Britain and The Netherlands. *Proc Biol Sci*. 2010;277(1683):933-42.

17. Jones RM, Brosseau LM. Aerosol transmission of infectious disease. *J Occup Environ Med.* 2015;57(5):501-8.
18. Alsved M, Widell A, Dahlin H, Karlson S, Medstrand P, Londahl J. Aerosolization and recovery of viable murine norovirus in an experimental setup. *Sci Rep.* 2020;10(1):15941.
19. Marshall JA, Bruggink LD. The dynamics of norovirus outbreak epidemics: recent insights. *Int J Environ Res Public Health.* 2011;8(4):1141-9.
20. Feeney SA, Armstrong VJ, Mitchell SJ, Crawford L, McCaughey C, Coyle PV. Development and clinical validation of multiplex TaqMan(R) assays for rapid diagnosis of viral gastroenteritis. *J Med Virol.* 2011;83(9):1650-6.
21. Pang XL, Honma S, Nakata S, Vesikari T. Human caliciviruses in acute gastroenteritis of young children in the community. *J Infect Dis.* 2000;181 Suppl 2:S288-94.
22. Lysen M, Thorhagen M, Brytting M, Hjertqvist M, Andersson Y, Hedlund KO. Genetic diversity among food-borne and waterborne norovirus strains causing outbreaks in Sweden. *J Clin Microbiol.* 2009;47(8):2411-8.
23. Wang P, Goggins WB, Chan EYY. A time-series study of the association of rainfall, relative humidity and ambient temperature with hospitalizations for rotavirus and norovirus infection among children in Hong Kong. *Sci Total Environ.* 2018;643:414-22.
24. Levy K, Hubbard AE, Eisenberg JN. Seasonality of rotavirus disease in the tropics: a systematic review and meta-analysis. *Int J Epidemiol.* 2009;38(6):1487-96.
25. Nguyen JL, Dockery DW. Daily indoor-to-outdoor temperature and humidity relationships: a sample across seasons and diverse climatic regions. *Int J Biometeorol.* 2016;60(2):221-9.
26. Quinn A, Shaman J. Indoor temperature and humidity in New York City apartments during winter. *Sci Total Environ.* 2017;583:29-35.
27. Nguyen JL, Schwartz J, Dockery DW. The relationship between indoor and outdoor temperature, apparent temperature, relative humidity, and absolute humidity. *Indoor Air.* 2014;24(1):103-12.
28. Marr LC, Tang JW, Van Mullekom J, Lakdawala SS. Mechanistic insights into the effect of humidity on airborne influenza virus survival, transmission and incidence. *J R Soc Interface.* 2019;16(150):20180298.
29. Kiulia NM, Hofstra N, Vermeulen LC, Obara MA, Medema G, Rose JB. Global occurrence and emission of rotaviruses to surface waters. *Pathogens.* 2015;4(2):229-55.
30. Boxman IL, Verhoef L, Dijkman R, Hagele G, Te Loeke NA, Koopmans M. Year-round prevalence of norovirus in the environment of catering companies without a recently reported outbreak of gastroenteritis. *Appl Environ Microbiol.* 2011;77(9):2968-74.
31. Quee FA, de Hoog MLA, Schuurman R, Bruijning-Verhagen P. Community burden and transmission of acute gastroenteritis caused by norovirus and rotavirus in the Netherlands (RotaFam): a prospective household-based cohort study. *Lancet Infect Dis.* 2020;20(5):598-606.
32. Lopman BA, Steele D, Kirkwood CD, Parashar UD. The Vast and Varied Global Burden of Norovirus: Prospects for Prevention and Control. *PLoS Med.* 2016;13(4):e1001999.
33. Park D, Shahbaz HM, Kim SH, Lee M, Lee W, Oh JW, et al. Inactivation efficiency and mechanism of UV-TiO₂ photocatalysis against murine norovirus using a solidified agar matrix. *Int J Food Microbiol.* 2016;238:256-64.

Figure legends

Figure 1. Weekly incidence of various gastrointestinal viruses, detected with real-time PCR, according to season across seven consecutive years 2010-2016 (n=7337). All agents except norovirus GI and adenovirus have a strong seasonal pattern with winter epidemics and almost no occurrence in the summer months. gg2=norovirus GII (turquoise), rota=rotavirus (green), gg1=norovirus GI (purple), sapo=sapovirus (orange), adeno=adenovirus (blue) and astro=astrovirus (red).

Figure 2. Weekly positivity rate of norovirus genogroup II infections (% left axis; blue bars) according to average weekly outdoor absolute humidity ($\text{g H}_2\text{O}/\text{m}^3$ right axis; thick red line) and relative humidity (% right axis; thin green line) across the seven-year study period (2010-2016). Dashed lines represent gaps between weeks with missing AH and RH data.

Figure 3. Weekly positivity rate of rotavirus infections (% left axis; blue bars) according to average weekly outdoor absolute humidity ($\text{g H}_2\text{O}/\text{m}^3$ right axis³; thick red line) and relative humidity (% right axis; thin green line) across the seven-year study period (2010-2016). Dashed lines represent gaps between weeks with missing AH and RH data.

Figure 4. Weekly positivity rate of norovirus GII positives from the last week of no positive samples to peak positivity rate (% left axis; blue bars) in relation to outdoor absolute humidity ($\text{g H}_2\text{O}/\text{m}^3$ right axis; red lines) across seven seasons (2010-2016). Dashed lines represent gaps between weeks with missing AH data.

Figure 5. Weekly positivity rate of norovirus GII positives from peak positivity rate to the last week of no positive samples (% left axis; blue bars) in relation to outdoor absolute humidity ($\text{g H}_2\text{O}/\text{m}^3$ right axis; red lines) across seven seasons (2010-2016). Dashed lines represent gaps between weeks with missing AH data.

Response to reviewers

Spectrum02433-22 Outdoor absolute humidity predicts seasonal variation of norovirus GII infection.

Reviewer #1:

We welcome all suggestions and have edited the paper accordingly.

Line 11: reword 'outdoor climate factors' with 'weather conditions'.

Answer: The wording has been changed as suggested by reviewer #1.

Line 12: replace 'between weather conditions and' with 'with'.

Answer: The wording has been changed as suggested by reviewer #1.

Line 14: delete 'agents causing viral' and add 'viruses' after 'gastroenteritis'.

Answer: The wording has been changed as suggested by reviewer #1.

Line 20: replace 'agents with 'viruses'.

Answer: The wording has been changed as suggested by reviewer #1.

Line 24: AH: spell out in full the first time used.

Answer: The wording has been changed as suggested by reviewer #1.

Line 35: would replace 'impact', here and in line 70 with 'association'

Answer: The wording has been changed as suggested by reviewer #1.

Line 86: is the assay specific for enteric (group F) adenoviruses or for all adenoviruses?

Answer: The assay is not specific for enteric (group F) adenovirus, it detects all adenoviruses. Group F (40/41) analysis is occasionally performed in specific cases.

Reviewer #2:

1. *The use of a linear model (univariate or multiple) is appropriate for an outcome variable that is a proportion (positivity rate) or count (number of cases), and generalized linear model should be used instead.*

Answer: We agree that linear regression might not be suitable when the type outcome is a count or a proportion, since the distribution of count or proportion data can be skewed. In the present study, we have checked the distribution of the residuals from the linear regression analysis. The distribution is close to normal and symmetric in most cases. Therefore, we decided, in collaboration with an experienced medical statistician, that it was appropriate to use linear regression in our analysis.

2. *Further, with the use of time-series data the impact of autocorrelation and for meteorological data, seasonality, need to be explored and likely incorporated into the model.*

Answer: We agree. Autocorrelation and seasonality may skew the data, which is better accounted for in a mixed model. We therefore checked our data by doing identical linear regression models on randomly selected subsets of measuring points (weeks) across all six seasons. The models created by these random measuring points gave the same directions

in coefficients, indicating a low impact of autocorrection and seasonality on the overall conclusions.

3. *There does not appear to be a clear strategy for determining which meteorological variables and lag time lengths for each variable should be included in each of the multivariable models.*

Answer: We agree that the presentation of variable selection was incomplete. This has been clarified in the Study design – Meteorological data section (Lines 109-116).

Determination of lag time lengths is described in detail in the Statistical methods section (Lines 131-136).

Response to reviewers

Spectrum02433-22 R2 Outdoor absolute humidity predicts the start of norovirus GII epidemics.

Reviewer #2 (Comments for the Author):

Thank you for your justifications of your statistical analysis. I still feel that linear regression is not appropriate based on your outcome measure, detection rates, since it is not a continuous variable. I would strongly consider using a generalized linear mixed model, so you can accurately model your outcome and account for the temporal correlation you have with your repeated measurements and seasonality. Your data is not independent and needs to be accounted for in the model.

Response to reviewer #2 comments:

We agree with the comments from reviewer #2. As the underlying data are counts with a temporal dependency we chose to use a generalized linear mixed model based on the Poisson distribution including time series components. When fitting the models without accounting for time dependencies the residuals were autocorrelated, especially for norovirus GII. Adding a single autoregressive term removed the autocorrelation in the residuals. The `auto.arima` function in the 'Forecast' R-package selected models containing a mix of autoregressive and moving average terms to model the data. When adding weather factors as regressors models with only autoregressive terms were suggested, indicating that a single autoregressive term, e.g. an AR1 model, would be sufficient to account for the temporal dependencies.

January 25, 2023

Dr. Magnus Lindh
University of Gothenburg
Infectious Diseases
Gothenburg 41346
Sweden

Re: Spectrum02433-22R2 (Outdoor absolute humidity predicts the start of norovirus GII epidemics.)

Dear Dr. Magnus Lindh:

Thank you for revising your manuscript. It is my pleasure to let you know that the manuscript has been accepted, and I am forwarding it to the ASM Journals Department for publication. You will be notified when your proofs are ready to be viewed.

Sincerely,

Gabriel Parra
Editor, Microbiology Spectrum
